# Waves Out of the Korean Peninsula and Inter- and Intra-Species Replacements in Freshwater Fishes in Japan

**DOI:** 10.3390/genes12020303

**Published:** 2021-02-21

**Authors:** Shoji Taniguchi, Johanna Bertl, Andreas Futschik, Hirohisa Kishino, Toshio Okazaki

**Affiliations:** 1Graduate School of Agricultural and Life Sciences, The University of Tokyo, 1-1-1, Yayoi, Bunkyo-ku, Tokyo 113-8657, Japan; taniguchi@ut-biomet.org (S.T.); hirohisa.kishino@gmail.com (H.K.); 2Department of Mathematics, Aarhus University, Ny Munkegade, 118, bldg. 1530, 8000 Aarhus C, Denmark; johannabertl@fastmail.fm; 3Department of Applied Statistics, Johannes Kepler University Linz, Altenberger Str. 69, 4040 Linz, Austria; andreas.futschik@jku.at

**Keywords:** competitive exclusion, East Asia, freshwater fish, intra-species replacement, island model, migration

## Abstract

The Japanese archipelago is located at the periphery of the continent of Asia. Rivers in the Japanese archipelago, separated from the continent of Asia by about 17 Ma, have experienced an intermittent exchange of freshwater fish taxa through a narrow land bridge generated by lowered sea level. As the Korean Peninsula and Japanese archipelago were not covered by an ice sheet during glacial periods, phylogeographical analyses in this region can trace the history of biota that were, for a long time, beyond the last glacial maximum. In this study, we analyzed the phylogeography of four freshwater fish taxa, *Hemibarbus longirostris*, dark chub *Nipponocypris temminckii*, *Tanakia* ssp. and *Carassius* ssp., whose distributions include both the Korean Peninsula and Western Japan. We found for each taxon that a small component of diverse Korean clades of freshwater fishes migrated in waves into the Japanese archipelago to form the current phylogeographic structure of biota. The replacements of indigenous populations by succeeding migrants may have also influenced the phylogeography.

## 1. Introduction

Inter- and intra-species interactions can influence biogeographical distributions [1,2,3]. Among many forms of biotic interactions, replacement among competing species that are mutually exclusive is presumed to be an important factor in biogeography [4]. Gutiérrez et al. (2014) [5] detected the interspecific competition on the biogeography of two mouse opossum species in South America by noting the dominant distribution of one species, whereas another species could potentially distribute. For humans, it has been reported that series of waves of man originated in Africa and propagated around the world. When these waves interacted with pre-existing populations, hybridizations and sometimes replacements occurred along the way in a phenomenon known as “out of Africa” [6,7].

While evidence for segregation is identified as genetic differentiation between geographical regions, phylogeographic evidence of intra-species replacements due to competition has not been extensively examined. Most evidence supporting the existence of species replacement is found in the spatial division and isolation of species, where the distributions of one species are surrounded by those of another [5]. This contrasts with conventionally observed fragmentation, where populations are both genetically and spatially isolated from each other. Because fragmented populations are small, they have large genetic differentiation. They do not comprise a monophyletic group in phylogeny, but are interspersed by the main population. Conversely, local populations that have been recently divided by competitors of a different clade are genetically homogeneous and comprise a monophyletic group (Figure 1).

The Japanese archipelago is located at the periphery of the continent of Asia. The archipelago landmass originally formed the eastern margin of the continent of Asia. After the back-arc of the archipelago opened to about 17 Ma, the northeastern half rotated counter-clockwise, while the southwestern half rotated clockwise. The current position was reached at about 14 Ma [8] and fused into the current Japanese archipelago at about 6 Ma. The boundary between these northeastern and southwestern masses is called the Fossa Magna (Figure 2a). The Japanese archipelago is elongated in a bow along a north–southwest axis. The mountains extending along this archipelago generate numerous short rivers that discharge separately into the ocean. The Sea of Japan is deep and has isolated the islands from the continent of Asia, except for narrow bridges at either end during periods of lowered sea level. These access points provide potential routes for genetic the exchange of freshwater fishes on the Korean Peninsula with those on the Japanese archipelago. In the Japanese archipelago, the bottom of the Inland Sea was above sea level in glacial periods, and paleo-river systems connecting surrounding rivers (Figure 2b) enabled gene flow [9]. Therefore, the current Inland Sea probably represented a likely dispersal route. Eastern dispersal was blocked after the uplift of Suzuka and Nunobiki Mountains at about 1 to 1.5 Ma.

Since the Japanese archipelago with backbone mountains has few plains, there are limited opportunities for rivers flowing through different plains to connect. Therefore, gene flow among freshwater fishes is extremely low, as salinity barriers separate neighboring rivers. Dispersal within freshwater systems is prevalent, while dispersal between them is rare, with watersheds and oceans representing migration barriers. Between-system dispersal can occur by floods in lower basins, stream capture, or the appearance of a larger freshwater system that connects rivers in periods of decreased sea level during glacial periods. Limited genetic exchange between rivers lowers the rates of species replacement. In the Quaternary, the distribution of biota was influenced by ice sheets [10]. In the northern hemisphere mid-latitudes, such as those throughout Europe or North America, it is difficult to trace biogeographic history beyond the most recent ice-sheet formation of about 10 Ka. However, the Japanese archipelago and Korean Peninsula were never covered by ice sheets during glacial periods [11]. Therefore, many taxa suitable for appraising the effects of competition on the distributions of genetic clades might exist in this region.

**Figure 2 genes-12-00303-f002:**
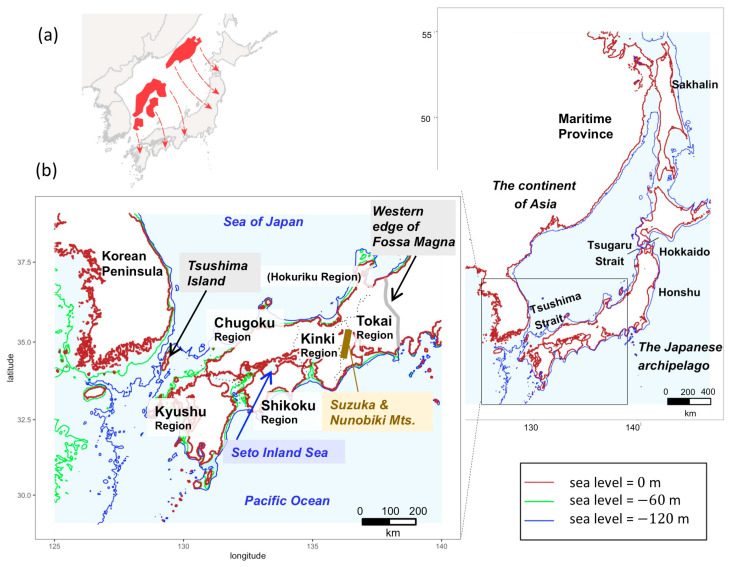
History of the Japanese Archipelago and study area. (**a**) Paleogeography of the Japanese archipelago inferred from geophysical information [12]. The red arrows indicate the movement of the land masses from the original position. (**b**) Study area map. An inferred map of the Far East during the last glacial period. The Seto Inland Sea, surrounded by Kyushu, Chugoku, Shikoku, and Kinki regions, with an average water depth of 38 m. Green and blue lines represent 60 and 120 m bottom depths [13,14], respectively, and allow for inferences to be made about coastal lines at times of lowered sea level.

A key factor in the establishment of the freshwater fish fauna of Japan is the isolation of these islands and their rivers, which affected the rates of expansion of migrant faunas. Here, we examine the effects of the migration and intra- and inter-species replacements on the phylogeographic structure in Japan. For this purpose, we analyze four examples of phylogeography in freshwater fish, *Hemibarbus longirostris*, dark chub *Nipponocypris temminckii*, *Tanakia* ssp. and *Carassius* ssp., whose distribution includes the Korean Peninsula and Western Japan. The first three examples have never been the target of commercial fisheries. Phylogenetic analysis combined with the geographical distributions of clades reveals that migrants from the Korean Peninsula have replaced indigenous populations in Japan. A simulation-based Bayesian analysis of *N. temminckii* reveals that the former dominated the latter significantly.

## 2. Materials and Methods

### 2.1. Sampling and Sequencing

Over 30 years, Toshio Okazaki (T.O.) amassed samples of freshwater fish from throughout Japan and Korea. From each sampling site, representative fishes were collected by various procedures, such as netting and angling. Fishes were fixed appropriately for molecular analyses (ETOH or −20 °C).

*H. longirostris:* samples were collected between 1990 and 2009 in Japan, and between 1991 and 1994 in South Korea. Among them, 27 individuals from 15 sites in Japan and 63 individuals from 20 sites in South Korea were subjected to analysis. By conducting PCR-RFLP analysis with 15 × 4-base pair recognition restriction enzymes (Appendix A), individuals were selected for sequencing. At each site, one individual was selected for sequencing when all individuals from a site had the same banding pattern. When there were different banding patterns among different individuals, then all such individuals were sequenced. To extract high resolution phylogeographic information, we chose the rapidly evolving mitochondrial gene NADH dehydrogenase subunit 2 (ND2) as a molecular marker. As a result, 42 × ND2 sequences of 584 bp were obtained.

*N. temminckii*: samples were collected between 1988 and 2013 in Japan, and between 1991 and 1994 in South Korea. Among them, 561 individuals from 340 sites in Japan and 93 individuals from 57 sites in South Korea were subjected to analysis. By conducting PCR-RFLP analysis with 13 × 4 base pair recognition restriction enzymes (Appendix A, individuals were selected for sequencing. At each site, one individual was selected for sequencing when all individuals from a site had the same banding pattern. Additionally, when individuals from many geographically close streams had the same banding pattern, individuals from equidistant streams were subsampled for sequencing. When there were different banding patterns among different individuals, then all such individuals were sequenced. As a result, 309 individuals from 248 sites in Japan and 41 individuals from 32 sites in South Korea were sequenced. We sequenced a partial region of ND2, 600 bp. To calculate the rate of ND2 molecular evolution, cytochrome b sequence data were obtained from 22 samples of individuals representing all clades (ranging from 1–6 samples per clade). Sequencing followed the procedures described in Appendix A.

Oily bitterling *T limbata*, *T. koreensis* and related species: samples were collected between 1989 and 2013 in Japan. Among them, 97 individuals from 47 sites were subjected to analysis. Samples of *T. koreensis* and related species were collected between 1991 and 1994 in South Korea. Among them, 17 individuals from 15 sites were subjected to analysis. As for genus *Tanakia*, individuals were screened for sequencing based on the PCR-RFLP analysis with 15 × 4-base pair recognition restriction enzymes (Appendix A). As a result, 70 × ND2 sequences of 741 bp were obtained.

*Carassius auratus*, *C.* sp. and Japanese white crucian carp *C. cuvieri* Temminck and Schlegel, 1846: samples were collected between 1989 and 2007 in Japan, and between 1991 and 1994 in South Korea. Among them, 274 individuals from 48 sites in Japan, and 101 individuals from 41 sites in South Korea were subjected to analysis. Three goldfish that we analyzed have been exploited commercially, unlike *N. temminckii* and *T. limbata*. Largely because of between-river transplantations, the PCR-RFLP analysis with 10 × 4-base pair recognition restriction enzymes (Appendix A) did not identify haplotypes that characterized regions. Therefore, all individuals with different banding patterns were sequenced. As a result, 154 × ND2 sequences of 600 bp were obtained. The obtained sequences were deposited in DDBJ/ENA/GenBank (accession numbers were LC566827- LC566870, LC566893- LC567047, LC567225- LC567298 and LC567958-LC568289).

### 2.2. Sampling Sites and Geomorphological Information

Precise collection sites were identified by T. O., based on field notes. Bathymetry was determined from ETOPO1 data [15] using R [16] marmap [17]. A shaded-relief map was made from the elevation chart [18], with marine areas assembled using data from the Hydrographic and Oceanographic Department, Japan Coast Guard. River data were obtained from the National Land Numerical Information download service [19].

### 2.3. Phylogenetic Analysis

A phylogenetic tree was constructed using the maximum likelihood (ML) method implemented in MEGA7 [20], and the Bayesian method implemented in BEAST 2.4.7 [21]. Bootstrap values of the ML-estimated branches were obtained by analyzing 1000 resampled alignment columns. To conduct MCMC analysis, we iterated 10,000,000 times and took samples every 1000 iterations.

We estimated the ancestral distributions of the clades by BayArea [22] and those of the OTUs by BayesTraits V3 [23]. BayesTraits analysis of the binary traits focuses on the change of range crossing the Tsushima Strait (Figure 2). On the other hand, BayArea penalizes long-distance dispersal with the distance-dependent dispersal-rate modifier. We set 21 discrete areas (Appendix A): 15 areas in Japan following Watanabe (1998) [24] and 6 areas in Korea which correspond to the major river systems. We randomly selected 1 sample from each clade in the phylogenetic tree, gave the presence/absence information of each clade over the 21 area, and estimated the geographical range of ancestors along the phylogenetic tree of each taxa. The phylogenetic trees of selected samples were also estimated by BEAST 2.4.7. All the parameters of the BayArea were set to default values.

For BayesTraits analysis, we extracted a 10% subsample of the MCMC sample of 9000 trees by sampling at regular intervals, in order to account for uncertainty in the phylogenetic tree. We used it as an input together with trait data (assignment of sampling site: Korea or Japan). The following subsections describe the details of the analysis specific to each species.

#### 2.3.1. *H. longirostris*

We used sequences of *H. mylodon* and barbel steed *H. barbus* as outgroups for *H. longirostris*. The Bayesian Information Criterion (BIC) selected the TN93+I model as the best model of nucleotide substitution. We confirmed that the phylogenetic clades of the current *H. longirostris* population in Japan were formed by a migration event from Korea.

#### 2.3.2. *N. temminckii*

The 309 sequences for *N. temminckii* included 109 unique sequences. *N. sieboldii* was used as an outgroup. BIC selected the TN93+I model as the best model of nucleotide substitution. In the Bayesian tree inference, we assumed TN93+I as the model of nucleotide substitution. The coalescent process model of a population in equilibrium was adopted as a prior for the tree. As a prior on the rate of molecular evolution, we assumed the log-transformed values followed an autocorrelated normal distribution [25]. The mean evolutionary rate was obtained by converting the frequently cited cytochrome b molecular evolutionary rate, 0.76% per site per MY [26], to that of ND2. By comparing the average evolutionary distance of cytochrome b sequences (0.0513) and corresponding ND2 sequences (0.0627), the ND2 evolutionary rate was estimated at 0.93%. We estimated divergence times in the Bayesian framework using the MCMCTREE package implemented in PAML 4.9 [27]. Detailed methods are described in Appendix A. We confirmed that the phylogenetic clades of the current *N. temminckii* population in Japan were formed by multiple migration events from Korea.

#### 2.3.3. *T. limbata* and Its Related Species

BIC selected the TN93+G+I model as the best model of nucleotide substitution of *T. limbata*. We used sequences of *T. lanceolate*, Tokyo bitterling *T. tanago*, *Acheilognathus rhombeus*, and big-scaled redfin *Tribolodon hakonensis* as outgroups for *T. limbata*.

#### 2.3.4. Genus Carassius

BIC selected the TN93+I model as the best model of nucleotide substitution of genus *Carassius*. We used sequences of *Cyprinus carpio* as the outgroup for *Carassius*.

### 2.4. Simulation-Based Testing of the Intra-Species Replacements

To test our hypothesis that indigenous populations of fishes were replaced by new migrants, we simulated the formation process for the clade distribution of *N. temminckii*. By contrasting simulated patterns in the distribution of this species under various competitive scenarios with the observed pattern, we examined if information contained in sequences and their sites provided evidence for intra-species replacement. The simulation assumed the timing of migration from Korea and dispersal rate within the Japanese archipelago, and the replacement rate among intra-species clades specifies the formation process. By estimating these parameters via approximate Bayesian computation (ABC) [28], the significance of the biogeographic evidence on intra-species replacement was tested.

As summary statistics, sensitive to model parameters [29], we extracted features from the geographical distributions of clades that contained information on the timing of migration, and dispersal and replacement rates. We adopted the combination of clade/nested clade distances in Nested Clade Phylogeographic Analysis (NCPA) [30,31] and spatial autocorrelation (Appendix A). NCPA was designed to associate the haplotype tree and geography through clade and nested clade distances, and to infer evolutionary history from population structure [32]. NCPA was very popular in the 1990s but became outdated since its evolutionary interpretation of the pattern of clade/nested clade distances was argued [33]. However, we adopted the clade/nested clade distance as summary statistics in our ABC simulation because we had hoped that it was sensitive to the spatial division and isolation caused by replacement. Spatial autocorrelation measures the correlation between genetic and geological distances and has a small value when a sample includes individuals that are surrounded by different neighboring clades. For a detailed explanation of the simulation algorithm, see Appendix A.

### 2.5. Data Availability

The obtained sequences were deposited in DDBJ/ENA/GenBank (accession numbers were LC566827- LC566870, LC566893- LC567047, LC567225- LC567298 and LC567958-LC568289). Sequence data and sampling sites are also available from the Appendix A. All freshwater fishes sampled by T.O., including *H. longirostris*, *N. temminckii*, *C. cuvieri* and *T. limbata*, are kept at Seikai National Fisheries Research Institute. Dr. Koichi Hoshino is responsible for the “Okazaki collection.”

### 2.6. Code Availability

All analysis scripts can be found in the Appendix A and at https://github.com/ShojiTaniguchi/Division_and_Isolation (accessed on 11 February 2021).

### 2.7. Ethical Statement

The guidelines or policies on conducting animal experiments are stated in “Act on Welfare and Management of Animals” enacted in 1973 revised in 1999 and “Standards relating to the Care and Keeping and Reducing Pain of Laboratory Animals” (Notice of the Ministry of the Environment No. 88 of 2006). Most of the specimens used for the analysis were collected before 2000, while T.O. was working for the National Research Institute of Aquaculture and National Research Institute of Fisheries Science, which also set up its guideline on animal experiment in 2008. However, fish is not subject to any of the above regulations or the guideline. Since there were no regulations or guidelines regarding the handling of fish body, the fish specimens were either immediately frozen or preserved in alcohol. The fish species we collected are commonly distributed and certainly not designated as nationally protected species or subject to any regulations.

## 3. Results

### 3.1. Phylogeography of H. longirostris

ML tree for *H. longirostris* identified three monophyletic clades (I, II, III). All individuals in clades I and III were sampled in South Korea, while clade II included individuals from South Korea and Japan (Figure 3). Clade I is distributed in the central region of the Korean Peninsula, and Clade III is distributed in the southeastern region of Korea. The distribution of clade II includes two isolated areas across the Tsushima Strait—the southeast region of Korea, and Western Japan, the latter distribution expanding from the Chugoku region facing the Seto Inland Sea (hereafter “Inland Sea”) to the western part of the Tokai region. The results of the BayesTraits (Figure 3) and the BayArea (Appendix A) indicated that *H. longirostris* was originated in Korea. After the split of clade II, the Korean distribution of clade II was limited to the southern area. On the other hand, clade II expanded its distribution into Japan. This indicates that the three clades diverged in Korea, with a clade splitting from the Korean population and migrating onto the Japanese archipelago via a land bridge.

Two land bridges between the continent and Japan are considered possible migration routes: the first between Korea and Western Japan, and the second between the maritime province of Siberia and Hokkaido in Japan, through to Sakhalin in Russia (Figure 2b). The deep Tsugaru Strait between Honshu (the main island) and Hokkaido represented a dispersal barrier for freshwater fishes [34]. Since their natural habitat does not extend to Eastern and Northern Japan, *H. longirostris* apparently used the land bridge to migrate from Korea to Western Japan.

### 3.2. Phylogeography of N. temminckii

ML and Bayesian trees for *N. temminckii* consistently identified seven clades (A–G, Figure 4a and Appendix A). The monophyly of clade F was significant in the Bayesian tree. All individuals in clades A, B, and D were sampled in South Korea, while clade C included individuals from South Korea and Japan (Figure 4b). The Korean population (C) is currently regarded as a distinct species, *N. koreanus*. Individuals in clade F were sampled in Japan, while those in clade G included individuals from both Japan and Korea. Clade F is distributed in the regions of Kinki, Chugoku, and Shikoku, whereas clade G is distributed in the regions of Chugoku, Shikoku, and Kyushu (Figure 4b). The distributions of the two clades appear to overlap in Chugoku and Shikoku, but fine-scale sampling in these regions reveals they are actually segregated (Figure 4c).

The distribution of *N. temminckii* in clade C includes two largely separated geographical regions—the southeast region of the Korean Peninsula, and the Tokai region of Japan. Clades F and G occupied areas west of the Tokai region [38]. Clade G is widely distributed throughout the southern region of Korea, and Kyushu, Chugoku, and Shikoku in Japan, whereas clade F is absent in Kyushu, but mostly present in Kinki (Figure 4b,c). A few samples of clade F were obtained in Tokai, at the upper reaches of the rivers running eastern side of Suzuka Mountains. Clade G expands around the Inland Sea, as does clade F, which is also found on the Sea of Japan side of Chugoku, and the Pacific Ocean side of Shikoku (Figure 4c), both isolated from the Inland Sea by mountains. In some areas, fishes in clade F occurred upstream or in mountain locations, while those in clade G occurred in the same river system in the mainstream or downstream (Figure 4d). The results of the BayesTraits (Figure 4a) indicated three main migration events into Japan; clades C, F, and G diverged in Korea, with some clades splitting from the Korean population and migrating onto the Japanese archipelago via a land bridge. For reasons similar to *H. longirostris*, *N. temminckii* apparently used the land bridge to migrate from Korea into Western Japan. BayArea (Appendix A) also indicated that *N. temminckii* was originated in Korea.

Since clade C inhabits the Tokai region, an area farthest from the land bridge in the Japanese distribution of *N. temminckii*, it represents the oldest clade in Japan (Figure 4b). Similarly, clade F is likely to be the second oldest clade, as it is distributed in the area second farthest from the land bridge. Clade G represents the youngest clade, as its distribution throughout Japan and Korea is closest. The existence of the Fossa Magna east of Tokai and Hokuriku (Figure 2b) might explain why *N. temminckii* did not expand further east. In glacial periods two paleo-river systems in the Inland Sea (Figure 4c) enabled gene flow [9]. Therefore, the current Inland Sea was a likely dispersal route. The consecutive migration of clades C, F, and G, followed by their dispersal might explain the present distribution of this species in Japan. Estimated divergence times are consistent with this scenario. As shown in Appendix A, node 1 (the split of clade C between Korea and Japan) is older than node 2 (the period of migration of clade F), with a posterior probability of 83.8%. Additionally, node 1 is older than node 3 (the period of migration of clade F from Korea), with a posterior probability of 98.2%. Although the phylogenetic tree does not provide a decisive chronology of nodal migration, the distribution map clearly suggests node 1 is older (Figure 4b). Bayesian tree estimation (Appendix A) reveals that migration of clade C occurred 1.52 Ma (0.876–2.04), clade F 1.31 Ma (0.896–1.74), and clade G 1.12 Ma (0.713–1.43). Therefore, we assume that the split in clade C between Korea and Japan occurred first, with clade F and G diverging from a common ancestor.

### 3.3. Significance of Intra-Species Replacement

The *N. temminckii* clades F and G are similar, with average nucleotide distances of 1.78%, far lower than genetic distances among salmonid species [39]. The taxonomic status of clade C is confusing; its population in Japan is regarded as the same species as clades F and G, while the population in Korea is considered to represent *T. koreensis.* The divided pattern of clade C is presumably at the intra-species level while the taxonomy requires review. Distributions of *N. temminckii* within Japan were simulated using various parameters (Figure 5a). The initial state was a universal distribution of clade C in Western Japan. Clades F and G migrated into Western Japan and expanded their distributions. Depending on replacement rate, resultant distributions differed (Figure 5b,c). To avoid the effects of differing sampling effort, we considered the dispersal of individuals on a lattice-like grid in Japan. Our model assumes that dispersal distance follows a gamma distribution; it has four parameters: r (Ma) for the timing of migration of clade G, m for the dispersal rate (km/Kyr), s (km) for the scale parameter of the gamma distribution, and *α* for replacement rate. The timing of the migration of clade F was set to 1.31 Ma, inferred using a Bayesian procedure (Appendix A). Under this model, *α* was estimated at 0.774 (0.554–0.951) (Appendix A), which is significantly higher than that for a neutral relationship (0.5). We, therefore, reject the null hypothesis of selective neutrality that assumes that the three clades C, F, and G have equal fitness. Of other parameters, *m* was extremely low, estimated at 0.345 (0.0135–0.860) km/Kyr, and *s* at 20.2 (5.33–40.1) km. The point estimates of *m* and *s* indicate that short dispersal was more frequent than long dispersal. The migration of clade G was dated at 0.862 Ma (0.552–1.30). The MCMCTREE gave a conditional credibility interval of 0.862 Ma (0.543–1.238).

The estimated migration time of clade G is mostly consistent with the Bayesian time estimation under a relaxed clock. The Bayesian time estimation by BEAST dates the timing of migration of clade G as 1.120 Ma (0.713–1.430). As the estimated timing of migration of clade F is 1.31 Ma, clade G migrated to Japan 1.013 Ma (0.696–1.263).

Clade F is widely distributed in Western Japan, but the distribution is isolated by Clade G in Chugoku and Shikoku. This pattern suggests that migration of clade G occurred after migration of clade F had settled into this habitat. Based on the fossil record, it is assumed that a land bridge between Kyushu and the continent of Asia formed at 0.43 Ma, 0.63 Ma, 1.2 Ma, and at about 5.3 Ma [40]. At the Last Glacial Maximum, sea-level was about 120 m lower than at present [14]. Since the current minimum water depth between Korea and Kyushu is 130 m, the glacial periods were not necessarily accompanied by land-bridge formation. While the migration of clade F and subsequent migration of clade G occurred in the Pleistocene, the exact timing of migration remains unresolved.

### 3.4. T. limbata and Related Species, and C. spp. and C. cuvieri

Three clades (1, 3, and 4) of *T. limbata* were consistently identified in Western Japan (Figure 6a). Clades (5–7) occurred in Korea, where they were classified as one of *T. koreensis*, *T. latimarginata* or Korean bittering *T. signifer*. Clade 2 includes *T. limbata* from a mountainous location of Japanese Chugoku, and *T. somjinensis* from the upper reaches of the Korean Seomjin River. *T. somjinensis* is geographically restricted to the upper reaches of the Seomjin River. *T. koreensis* is distributed in the southwestern Korean Peninsula, including the middle to lower reaches of the Seomjin River [41].

The result of the BayArea (Appendix A) showed the geographical origin of all the clades (1–7) located in Korea, and the main distribution area of clades (1–4) changed to Japan. The Korean distribution of this group became restricted to the Seomjin River, which is now the habitat of *T. somjinensis*. On the other hand, the BayesTraits showed a different process. The common ancestor of clades 1–6 distributed from Korea to Japan (posterior probability of Japan: 53.2%). Then, clades 5 and 6 established their distribution in Korea. On the contrary, the common ancestor of clades 1–4 once established its distribution in Japan (posterior probability of Japan: 97.3%), and *T. somjinensis* migration from Japan. The difference in ancestral state reconstruction is probably due to the difference in the topology of the trees. Since the order of divergence of the clades 1–4 were not highly supported, the subsampling of the sequences can change the topology.

The fact that clades 1–4 comprised a large monophyletic group and the habitat of *T. somjinensis* is restricted indicates their past continuous distribution from Korea to Japan and isolation by *T. koreensis* afterwards. Therefore, *T. koreensis* expanded its distribution in Korea later than *T. somjinensis*. However, the geographical information contradicts the result of BayesTraits, which indicated that the *T. somjinensis* newly arrived from Japan. Since it is difficult to expect that the ancestor of *T. somjinensis* went up the Seomjin River where *T. koreensis* was dominant, the result of BayArea seems to reflect the reality. Further examination may be necessary.

*Carassius cuvieri* is endemic to Lake Biwa, Japan, and is represented by clade b in Figure 6b. Since *C. cuvieri* was artificially transplanted to various places in Korea and Japan, the same sequences as the original population in Lake Biwa were found in Korea. However, in the Korean region of Haepyeong, we detected another sequence, which we presume is indigenous unless the same sequence occurs in Lake Biwa; the fish differs morphologically and had initially been identified as an individual from Korean clades. As transplanted *C. cuvieri* are morphologically indiscernible from *C. cuvieri*, it is unlikely that these Haepyeong individuals were transplanted. It is possible that clade b was previously continuously distributed in Western Japan and Korea and that it has since become isolated by other clades in Korea. BayesTraits indicated that the common ancestor of the clade b was Korea (posterior probability of 88.4%). BayArea analysis of genus *Carassius* did not converge.

## 4. Discussion

The Japanese freshwater fish populations we examined were derived from one or few clades of Korean populations. Other vertebrates in Japan, such as the Siberian weasel *Mustela itatsi* and the Japanese tree frog *Hyla japonica* derived from one of a few clades of the continent as well. These phenomena suggest that the geographical origin of Japanese taxa is from Korea, and the migration waves out of Korea should be key factors in their distribution patterns if the Japanese taxa have related ones in Korea (Appendix A). The effects of migration might differ among taxa. In the case of *H. longirostris*, migration occurred only once, and the homogeneous genetic structure of this species in Japan indicates that dispersal occurred over a short period of time. The distribution of *H. longirostris* expands around the Inland Sea. On the contrary, *N. temminckii* migrated into Japan several times, probably resulting in the spatial structure of clades in Japan. The intermittent formation of the land bridge and stochastic success of migration through the land bridge generated taxon-specific waves into Japanese archipelago.

Genetic data from samples of multiple freshwater fishes from the Korean Peninsula and Japanese archipelago suggest waves of migrations of species from Korea have established themselves in Japan. Previous studies have interpreted the genetic structure of Japanese freshwater fish to have been caused mainly by vicariance. For example, the uplift of Suzuka and Nunobiki Mountains in the early Pleistocene (Figure 4c) [42] caused genetic differences in fishes between the Tokai and Kinki regions [43]. However, by incorporating phylogeographic data from Korea, we demonstrate that genetic differences in *N. temminckii* across Suzuka and Nunobiki Mountains are derived from different migration events from Korea, with the divergence of the two clades dating to the Pliocene (6.15 Ma; Appendix A) being significantly older than the uplift of the mountains. We also found older clades to be discontinuously distributed, separated by newer migrants. Such distributions cannot be fully explained by processes of diffusion or vicariance [44], which predict genetic differentiation among components of division (Figure 1). If geographically divided individuals of older clades are genetically similar, their distribution was formerly continuous until recently because gene flow is extremely low in freshwater fishes. Extinction events and the expansion of distributions of other clades produced existing patterns.

Our hypothesis, that intra- and inter-species replacement has occurred in the process of successive migrations of taxa from Korea, can explain the discontinuous distributions of freshwater fish taxa in Japan. The simulation suggests that the effects of replacements are significant compared to the null model of neutrality. In addition, divisions between clades exist in areas where dispersal barriers exist. For *N. temminckii*, the distribution boundaries occur around uplifted mountains (Suzuka and Nunobiki Mountains; Figure 4c) and the east–west axis of mountains in the Chugoku and Shikoku regions (Figure 4c,d). East of the Inland Sea, the uplifted Suzuka and Nunobiki Mountains presumably acted as barriers to dispersal for newer migrants, in addition to preventing replacement of an older clade (Clade C) in the Tokai region. A few samples of *H. longirostris* from the Tokai region probably exist because of artificial transplantation [45], as sequences from these individuals are identical with samples from the Kinki region. Therefore, the eastward dispersal of *H. longirostris* was stopped in the same way as it was for *N. temminckii*. The most recent migrant (clade G) had a continuous distribution around the Inland Sea, with an older clade (F) found in rivers discharging into the Sea of Japan, the Pacific Ocean, and in the upper parts of rivers flowing into the Inland Sea. The dispersal of the newer migrant probably caused replacement, with the old clade divided into an area where the former could not reach.

The formation of refugia might also contribute to discontinuous distributions. Many terrestrial animals in Japan sought refugia during glacial periods by migrating south or to low-altitude areas, and expanded their distributions again during interglacial periods [46]. However, dispersal opportunities between rivers for freshwater fishes are more limited; they can either overwinter in local springs [47] or go extinct. Therefore, we postulate that the main factors contributing to the present-day distributions of Japanese freshwater fishes are not the distributions of climatic refugia, but divergence, dispersal [48,49], migration and replacements.

The signature of replacements is also found in other freshwater fishes in Japan. For example, Japanese rice fish *Oryzias latipes* in Western Japan have a divided distribution [49,50], and the pike gudgeon *Pseudogobio esocinus* (Temminck and Schlegel, 1846) [9,51] and Japanese spined loach *Cobitis biwae* Jordan and Snyder, 1901 [48] have older clades in eastern and parts of western Japan, and more recent clades in wide areas of western Japan. Our hypothesis may not be restricted to freshwater fishes, as the distributions of closely related moles, *Mogera wogura* and *M. imaizumii*, are also similar to that reported for *N. temminckii*, with *M. wogura* expanding from Korea to western Japan and *M. imaizumii* in eastern Japan and some isolated area in western Japan (Appendix A) [52,53]. These distributions are parapatric, with the latter having been replaced by the former at the distribution boundary; replacement of *M. imaizumii* by *M. wogura* was considered a decisive factor in the formation process [54].

An interesting problem is whether replacements are confined to mitochondrial genomes, or if they also extended to nuclear genomes. Allozyme analysis of *N. temminckii PEPA* nuclear locus revealed two-allele polymorphism at the upper portions of a few rivers located in the western Tokai region. In the Tokai region where clade C was fixed, the allozyme haplotype was fixed (*120) except the above populations. In the Kinki region where clade F was fixed, the allele is fixed to another (*100). The observed number of individuals by genotype are in Hardy–Weinberg equilibrium. The haplotypes of mitochondria and allozyme were not consistent, which means that the individuals over the boundary crossed randomly. They were probably caused by stream capture (Appendix A) [38]. Since clades F and G are genetically much closer than clades C and F, they could also hybridize. This implies that replacements occurred through a change in the composition of admixture. In the case of rosy bitterling *Rhodeus ocellatus*, when a non-native subspecies from the continent of Asia (*R. o. ocellatus*) was introduced into a population of Japanese native freshwater *R. o. kurumeus*, the mitochondrial and nuclear DNA of the former was replaced by the latter through hybridization [55]. According to Ohta (1972) [56], the efficiency of selection is negatively correlated with population size. The Korean Peninsula is part of the continent of Asia, and continental populations might have experienced higher competition than Japanese ones, and accordingly have higher fitness. Since newer migrants might have experienced higher selection pressures on the continent for a longer time, they may have higher fitness than indigenous species, resulting in the older clade being replaced by the newer clade. Japanese giant salamanders *Andrias japonicus* (Temminck, 1836) [57] and Japanese weasel *Mustela itatsi* Temminck, 1844 [58] are being replaced by related species that have been artificially transplanted from Korea or the continent of Asia.

Intra- and inter-species replacements have also occurred in Korea, where Korean *C. cuvieri* at Haepyeong occurred upstream of the Nakdong River, surrounded by the sister clade, *C. auratus*. This pattern is similar to the Korean *T. somjinensis*, which comprises a monophyletic clade (Clade 2) with Japanese *T. limbata*, and inhabits the upstream waters of Seomjin River, surrounded by *T. koreensis* and *T. latimarginata*. Local distribution barriers, such as waterfalls or flashy streams, may have helped these isolated clades survive, or survival may be due to chance. In Southwestern Korea, we sampled populations of *H. longirostris* (Clade II), comprising a monophyletic clade with the Japanese clade, which were geographically surrounded by other clades. In the case of *N. temminckii*, the clade from southwestern Korea (Clade C) also had a patchy distribution around this region. The Japanese tree frog *H*. *japonica* (Appendix A) and Siberian weasel *M*. *itatsi* (Appendix A) [59] have divided distributions of certain clades on Tsushima Island, a small island between Korea and Japan in the middle of the migration route, with populations in the middle of the Korean Peninsula or Russia, comprising monophyletic clades with those on Tsushima. Other clades were sampled in between (e.g., southern Korea). In these two species, we expect that the latter clades replaced the former clades in Korea, but the latter could not migrate to Tsushima Island. Both *T. somjinensis* and *N. koreanus* are regarded as Korean endemic species, but they are components of monophyletic clades with different species in Japan. The taxonomy of these species requires review.

While the waves of migration out of Korea are essential factors in phylogeography, other factors also need considering. For example, *C. cuvieri* migrated into Japan in a single wave because its population is monophyletic. However, the distribution of this species is restricted to the freshwater system of Lake Biwa; as it prefers calm waters, the loss of paleo-river systems around the Inland Sea with rising sea level may have reduced suitable habitat. A similar pattern was detected for the freshwater fish three-lips *Opsariichthys uncirostris* and related species [60]. The population of *T. limbata* in Japan was segregated into four clades. Among them, only the single clade (clade 2) contained the Korean population. To better understand the phylogeographic structure, a more comprehensive model is necessary. In this study, we show that the replacements consistently explain the division and isolation found in many taxa. For the further study, it is indispensable to set other hypotheses, such as the “homogenization of recent migrants,” and to evaluate them by comparing to the replacements. We analyzed one mtDNA locus and one nuclear DNA locus. Genome data would provide more information to assist with understanding the history of these taxa in Korea and Japan. Nevertheless, our hypothetical process involving waves of migrations and replacements can be applied for other places and for different taxa, such as the brown bear *Ursus arctos* (Appendix A) [61,62,63] and divided distribution of the mountain hare *Lepus timidus* Linnaeus, 1758 [64] in Hokkaido. Chinese rice fish *Oryzias sinensis* had a divided distribution in Korea, which might be replaced by a clade migrating from the Continent of Asia (Appendix A) [65]. Although our findings owe much to the suitable geographical conditions of the Japanese archipelago and Korean Peninsula, waves of migrations and replacements are the potential factors which may be more common and have more widely influenced the formation of biota than previously recognized.

## Figures and Tables

**Figure 1 genes-12-00303-f001:**
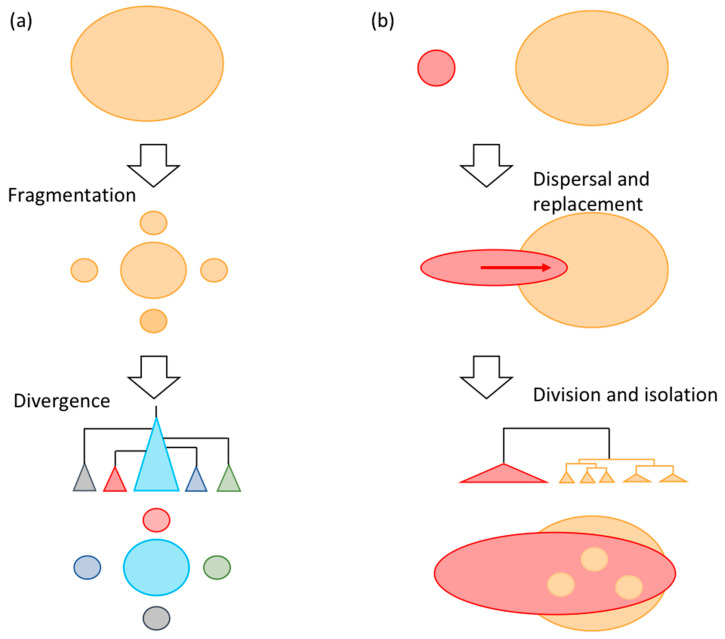
Schematic illustration of conventional fragmentation and replacement. (**a**) Conventional fragmentation, where fragmented populations diverge into small populations with unique genetic features. Isolated populations do not comprise monophyletic groups in the phylogeny but are interspersed by the main population. (**b**) Local populations (three small circles and upper and lower fragments colored by orange) recently divided by competitors from different clades are genetically homogeneous and comprise monophyletic groups.

**Figure 3 genes-12-00303-f003:**
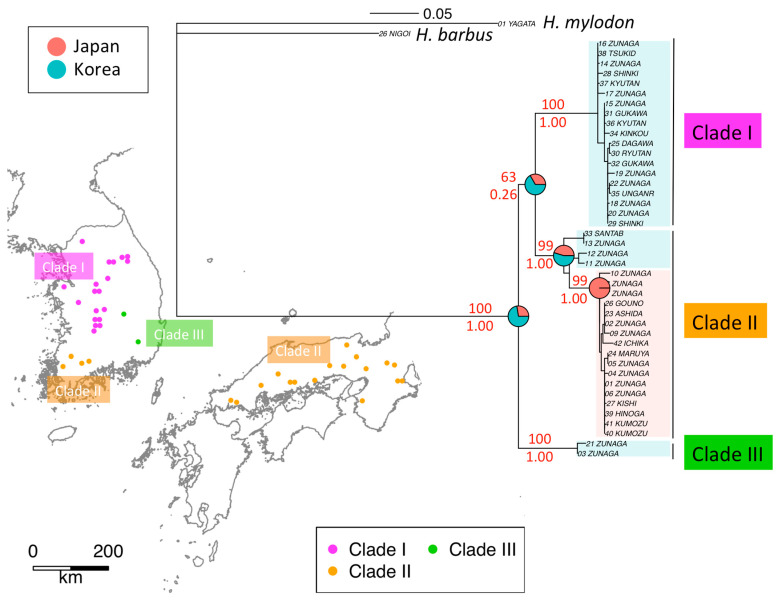
Phylogenetic trees of ND2 sequences and geographic locations of *H. longirostris* reconstructed from ND2 sequences. Red numbers in the phylogenetic tree indicate bootstrap probabilities (%) on the top and posterior probabilities on the bottom. Pie charts on the phylogenetic tree represents the Bayesian assignment of ancestral nodes to Japan and Korea.

**Figure 4 genes-12-00303-f004:**
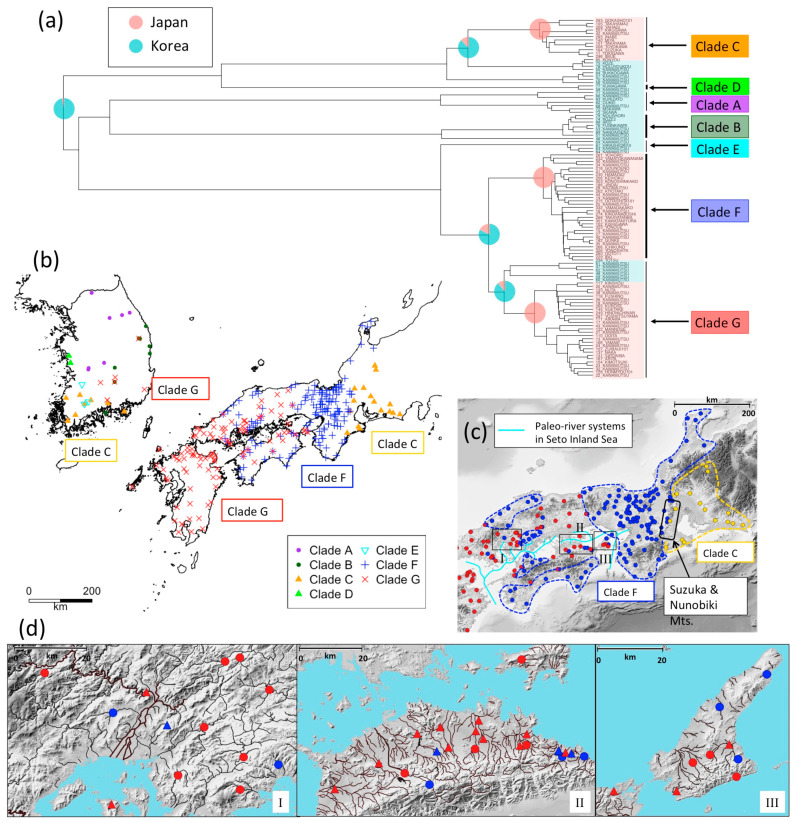
Phylogeny and distribution of *Nipponocypris temminckii*. (**a**) Bayesian assignment of ancestral nodes to Japan and Korea; (**b**) sample sites by clade; (**c**) distribution of clades on shaded-relief map; (**d**) focused distributions at sites I, II and III. Detailed distributions of individuals for which the type of a clade has been determined by PCR-RFLP analysis are displayed as triangular points (**d**). Colors represent the cleavage types of BstUI, DdeI, and TaqI that correspond to clades F and G (see Appendix A). Paleo-river systems (**c**) modified from original maps [35,36]. At localities I and III, clade F was sampled at sites surrounded by mountains, but at locality II, clade F was sampled upstream and clade G downstream. We used R [16] package GGTREE [37] for visualization in (**a**).

**Figure 5 genes-12-00303-f005:**
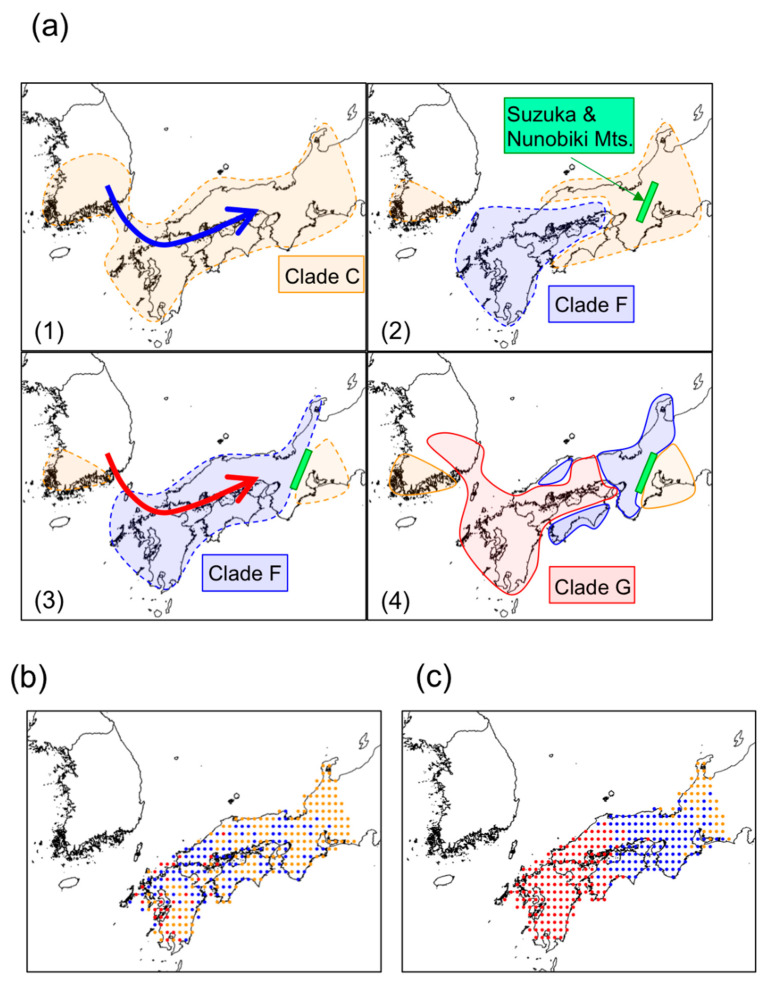
Schematic view and scenario of the formation process. (**a**) scenario with replacement for *Nipponocypris temminckii*: The number (1–4) at the bottom of each window represents the sequence in time and migration. The uplift of Suzuka and Nunobiki Mountains occurred in this process (2). (**b**,**c**) are examples of simulated *N. temminckii* distributions. (**b**) is the neutral scenario (m = 0.5, s = 20.2, α = 0.5, r = 0.853), and (**c**) is the best scenario with replacements (m = 0.345, s = 20.2, α = 0.774, r = 0.853). In the neutral scenario, the distribution of new migrants would expand where native populations had existed, and several clades would mix in a wide area. Yellow color represents clade C, blue represents clade F, and red represents clade G.

**Figure 6 genes-12-00303-f006:**
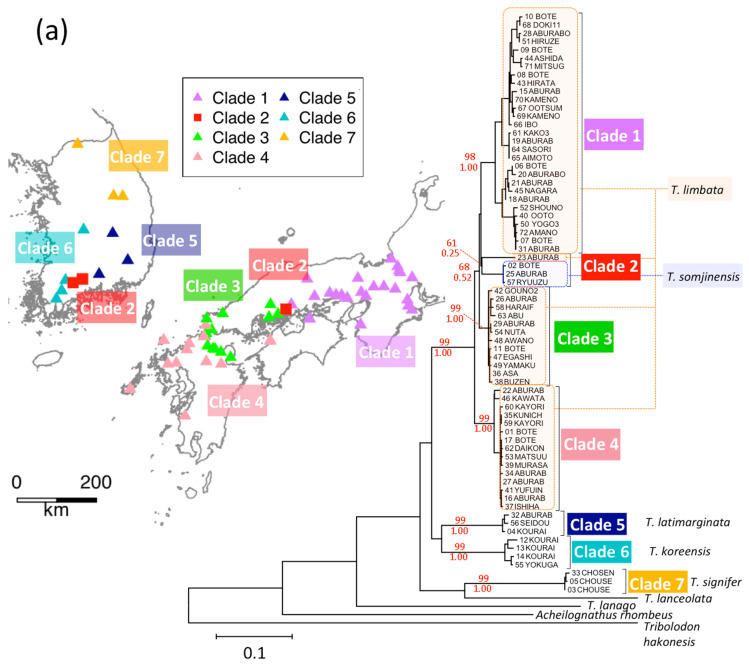
Phylogenetic trees of ND2 sequences and geographic locations: (**a**) *Tanakia limbata* and related species, (**b**) *Carassius* spp. As for *Carassius cuvieri*, the haplotype from Lake Biwa was found at a southern site in Korea (red rectangle). The haplotype from Haepyeong (red star) was very close to the haplotypes from Lake Biwa, but differed morphologically (it was initially identified as an individual from Korean clades c, d, and e).

## Data Availability

The data presented in this study are openly available. The obtained sequences were deposited in DDBJ/ENA/GenBank (accession numbers were LC566827- LC566870, LC566893- LC567047, LC567225- LC567298 and LC567958-LC568289). Sequence data and sampling sites are also available from the Appendix A. All freshwater fishes sampled by T.O., including *H. longirostris*, *N. temminckii*, *C. cuvieri* and *T. limbata*, are kept at Seikai National Fisheries Research Institute. Dr. Koichi Hoshino is responsible for the “Okazaki collection.”

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
