# Peer review of "Waves Out of the Korean Peninsula and Inter- and Intra-Species Replacements in Freshwater Fishes in Japan"

_genes, 2021, doi:10.3390/genes12020303_

Round 1

Reviewer 1 Report

The article “Waves out of the Korean Peninsula and inter- and intra-species replacements in freshwater fishes in Japan” does an impressive job of examining the biogeographical patterns of freshwater fishes in East Asia. Taniguchi et al. use four clades of freshwater fishes to show a strong pattern of migration (and in some instances repeated migration) from Korea to Japan. Additionally, they provide a nice set of evidence to refute the idea that geological processes are solely responsible for the contemporary distribution of Japan’s fishes but instead repeated migration has shaped their biogeography. I very much enjoyed reading this paper and think it will be of interest to a wide range of readers. I particularly found the Nipponocypris example very fascinating. I do have a number of comments that I think can be addressed with ease.

Major Comments:

L:30-32 This first paragraph does not do a good job setting up the paper. The first two sentences are fine, but the human example seems misplaced in its current context. Adding another 1-2 examples and a more direct statement to tie it into the main idea in the paper seems necessary.

L133: Regarding: “chrome b sequence data were obtained from 22 samples of individuals from all clades.” I assume you mean “… 22 samples of individuals representing all clades”. If this is the case, please reword and add a range of number of samples from each clade (or at least a range).

“… 22 samples of individuals representing all clades (ranging from # - # individuals per clade)”. It would also be worth specifying the individuals in the supplement.

L 378-390: I do not understand “suggests the T. limbata lineage formerly had a continuous distribution from Korea to Japan, but that this has since been divided by T. koreensis and T latimarginata.” Please rewrite this.

It seems like you are suggesting clade 1-4 had a continuous range and then clade 1,3, and 4 all independently split off. It is hard to make any supported conclusions, since you do not have the more robust analyses as you did with Nipponocypris, which is fine. Tanakia shows a phylogeographic patter opposite of Nipponocypris, with the newest clade being the most western group. Could an explanation be that a linage of the (T. koreensis and T. latimarginata) group migrated to Japan and speciated to form T. limbate. Followed by a migration back to Korea by a linage forming T. somjinensis? This seems like a more parsimonious explanation.

Figure 4: Topology of the clades in figure 4 is not the same as that in fig S2 or Fig S3. That should be addressed by explaining where the different trees come from. I may have overlooked this, but when looking at the dates in the sup figures I did not see any explanation and assume they you are using the optimal Bayesian tree? This does not alter any of the result and does not interfere with your conclusions, but I think it needs to be addressed.

Supplement L19: When you say “Following Hall & Nawrocki (1995), we carried out PCR…” I assume the PCR parameters are explained in the reference. What about the protocol for the restriction enzymes? Please include the protocol or cite the appropriate reference.

Minor Comments:

L40: You use the word “variation” but I think a better word might be differentiation if you are talking about the genetic variation between population. Based on your statement I read it as:

‘Because fragmented populations are small, they have large genetic variation within each population’

but I think you are trying to express the idea that between population there is a lot of variation.

L51: You may want to specify that this occurs in small neighboring rivers that are primarily coastal or on islands. In the Nile, Amazon, Mississippi, and Yangtze rivers there may be very high levels of gene flow across a large special area.

Figure 2b: A suggestion, but not necessary as it might look bad, I leave it up to you to decide. Can you shade in the land so the reader can have a little better perspective of where the different sea levers are with respect to modern-day Japan? I understand this might not be possible depending on how you created the image.

L91: I recommend changing “we analyze the phylogeography of four freshwater fish taxa” to something like “we analyze four examples of phylogeography in freshwater fish” since you are actually looking at more than four taxa. Likewise, on Line 94 you can change it to “The first three examples”.

L99-105: I would move this to become the 3rd paragraph. It seems like an odd place to make the broad comparison after you already addressed your specific objectives.

L118: Do you mean that when there were different banding patterns then all individuals were sequenced? Or that one individual representing each of the banding patterns was sequenced? Please specify.

L121: Remove “We sequenced the same 121 rapidly evolving ND2 for other freshwater fishes.”

L303: Clades C, F and => Clades C, F, and

I would recommend using the oxford comma (sometimes you use it like on L323), but either way stay consistent.

Again L315

L367: “Furthermore, not all migration events contributed to the endemicity of Japanese land mammals (Sato, 2017).” Seems a little out of place and the wording is not clear.

L373: “Sister clades (5–7)” => “Clades (5–7)”

5 and 6 are sister clades, but 7 is not sister to clade 5 and 6.

L374: Kim, Jeon & Suk, 2014 => (Kim, Jeon & Suk, 2014)

L402: Pallas, 1773. => (Pallas, 1773)

L402: “Other vertebrates in Japan such as the Siberian weasel Mustela itatsi Pallas, 1773 and the Japanese tree frog Hyla japonica (Günther, 1859) derived from one of a few clades of the continent as well.” Needs a citation. Or at least refer to the figures S5-S8.  

L475: “The observed number of individuals by genotype are consistent with the Hardy-Weinberg equilibrium.”

=>

“The observed number of individuals by genotype are in Hardy-Weinberg equilibrium.”

L579: Chen, Uwa & Chu, 1989 => (Chen, Uwa & Chu, 1989)

Supplement L27: When you say the same primer as that for H. longirostris was used, are you are talking about the ND2? Please specify.

Reviewer 2 Report

The submitted manuscript examined phylogeographic patterns in four freshwater fish species spanning the Korean Peninsula and drainages of western Japan using a single mitochondrial gene (ND2), and a combination of phylogenetic inference and phylogeographic reconstruction. I found the manuscript to be well-written, and with interesting results. The authors found concordance across taxa in a Korean origination, with evidence suggesting varying numbers of subsequent colonization events into Japan. While I question the method by which ancestral range reconstruction was performed (see below), the ‘multiple colonization’ hypothesis seems relatively solid. The authors however also claim a pattern of successive demographic replacement coinciding with these migration ‘waves’; the analytical support for this I find to be much less supported, and on weaker analytical grounds. Thus, I recommend the authors either dampen purported support for this scenario, or bolster analysis in its favor.

Here I will additionally summarize what I see as the major issues for the submitted manuscript; note that I’ve provided some more detail, as well as a more extensive list minor issues in the section below.

  1. Methods needs re-organization and more detail in several places. More details in the section below, but in many cases there is insufficient detail to fully judge what has been done, much less to actually replicate their analyses.

  1. Several of the important analyses were performed using out-dates analytical methods, or those which are of dubious applicability. I suggest the authors explore alternative methods for the phylogeographic reconstructions (BayesTraits and NCPA).
  • In particular I’m not certain that the ancestral range reconstruction was done appropriately. I’ve tentatively recommended that the authors re-examine using an explicit geographical model. However, there really isn’t sufficient detail to evaluation what exactly the authors have done (see comments re: methods below). As it is, it appears that the authors have used a model of Brownian motion, where “range states” (e.g., Japan or Korea) are treated as a discrete trait, with states being mutually exclusive. However, a real range may of course encompass multiple areas (as the author’s own Figure 1b shows). I suggest the authors either re-examine using an alternative model which has been explicitly developed with processes driving ranges in mind (e.g., dispersal/ vicariance) or provide a thorough justification of their methodological decisions.
  • Why was the range reconstruction method applied to only 2 of the 4 species?
  • Additionally, the NCPA procedure has fallen out of favor due to a propensity towards false-positive results (some citations below). Given this, I again suggest that the authors bolster both the level of detail in their Methods, as well as provide some discussion at the very least of caveats surrounding this approach.

  1. With the exception of Figure 1, the figures are not legible. Fig. 2: I suggest darkening the lines representing the coast lines, and if it is not necessary to have three separate depth contour lines, remove them. Fig. 3, 4a, 6a-b phylogenetic tip labels are way too small. Several of the maps are very difficult to read – consider removing any detail which is not pertinent so as to make them easier to digest at a glance. For example, in Fig. 4d is it necessary to have the DEM layer included? If not, all it does is make the map messier and more difficult to read.

Minor Issues

L20-21: The comparison with human ‘out-of-Africa’ seems rather unnecessary here.

L154: How did you edit and align the sequences?

L164-190: When I hit sections 2.3 and 2.4 I was thrown off, because the abstract and section 2.1 led me to believe I would see a comparative analyses across four species, yet these sections only detail phylogenetic methods for two. If analytical treatment differs across the species you are examining, I would suggest that this may be a reasonable place to make the nature of those differences (and the reasoning behing them…) clear. You may consider re-arranging to make Section 2.3: Phylogenetic Analysis, with sub-sections (2.3.1, etc) for each study species only having the necessary details which are specific to that species. You could then remove the redundancy e.g. in describing the methods of tree construction (MEGA7 and BEAST), nucleotide substitution model selection, etc. As it is I see no reason to repeat this information for each species, given that it doesn’t seem to differ.

L171/ 175: This is not enough information to understand how you performed ancestral range reconstruction. BayesTraits does not rely on a model of range evolution but treats any form of discrete character as evolving via Brownian motion A more appropriate analysis would employ one of the several explicit biogeographic models which have been developed. I believe there are also geographical models (using lat/ long) as an option in BayesTraits, which you might look into.

L172 and L180: Please provide the MCMC parameters. How many total iterations, how often were samples taken (every 100th or 1000th iteration?), etc. Was this done the same for both datasets?

L174: What sort of trait data?

L195-198: Unclear  

L201: Explain ‘formation process’ or re-word to make it clear what you mean

L204: “Assumed” (past-tense)

L207: How did you perform the ABC? Priors? There is a lot of information missing here.

L209-218: Not enough detail

L212: Hasn’t NCPA been shown to be unreliable?

http://dna.ac/filogeografia/PDFs/NCPA/Petit_08_Death_to_NCPA.pdf https://academic.oup.com/sysbio/article/59/4/415/1659579

L211: What Templeton statistics? This is the first mention

L220-226: I suggest you explain analyses for all species together – explain all phylogenetic analyses in Section 2.3, etc. As it is, this feels like the information in 2.5 was ‘tacked on’.

L228: I assume an NCBI accession or bioproject number will be placed here?

L233: Referenced repository does not exist.

L249: “mt tree” -> “tree”; also make this change throughout (no need to qualify as “mt” tree unless you are e.g. comparing with a “non-mt” tree, as it was made clear in methods that you are examining mtDNA)

L261 (and elsewhere): Phylogenies were produced using multiple methods, correct? Why then do Figures 3, 4, and 6 only provide a single set of results? If topologies were consistent between the Bayesian and ML phylogenies, indicate this in the Figure headings and provide nodal supports for both methods. E.g., 99/0.98; with ML bootstrap on top and Bayesian posterior probabilities on bottom.

L347 (and throughout): I would suggest presenting the phylogenies as time-scaled, with error bars at nodes representing the credibility intervals for the dates.

L514-515: Though is it possible that earlier migrations were ‘replaced’, or with (mitochondrial) genetic diversity being ‘hoomogenized’ by high levels of gene flow from more recent migrations?

Round 2

Reviewer 2 Report

I find the response and revisions by the authors to be sufficient and have no further comments.